# Peritoneal Oxygenation as a Novel Technique for Extrapulmonary Ventilation; A Review and Discussion of the Literature

**James R. M. Colbourne** [1,2,*] , **Khaled H. Altoukhi** [1] **and David L. Morris** [1,3]

1   Department of Surgery, St. George Hospital, Sydney 2217, Australia
2   Westmead Hospital, University of Sydney, Sydney 2145, Australia
3   St. George and Sutherland Clinical School, University of New South Wales, Sydney 2217, Australia
*   Correspondence: j.rmcolbourne@gmail.com; Tel.: +61-2-9133-2590

**Highlights:**

**What are the main findings?**

- Peritoneal oxygenation has significant potential for treatment of acute respiratory distress syndrome patients.
- It could offer an effective treatment for rescue ventilation in critically unwell patients.
- Further large animal studies are needed with further refinement of currently available techniques.

**What is the implication of the main finding?**

- We propose the further development of techniques and animal models with an eventual goal of evaluation in humans.
- Peritoneal oxygenation is a promising alternative to rescue ventilation is cases of extreme hypoxia.

**Abstract:** The COVID-19 crisis has highlighted the difficulties that might occur when attempting to oxygenate patients who have suffered a severe pulmonary insult, including in the development of acute respiratory distress syndrome (ARDS). Traditional mechanical ventilation (MV) is effective; however, in severe cases of hypoxia, the use of rescue therapy, such as extracorporeal membrane oxygenation (ECMO), may be required but is also associated with significant complexity and complications. In this review, we describe peritoneal oxygenation; a method of oxygenation that exploits the peritoneum's gas exchange properties in a fashion that is similar to peritoneal dialysis and has shown considerable promise in animal models. We have conducted a review of the available literature and techniques, including intraperitoneal perfluorocarbons, intraperitoneal jet ventilation, a continuous low-pressure oxygen system (PEROX) and the use of phospholipid-coated oxygen microbubbles (OMBs) through peritoneal microbubble oxygenation (PMO). We conclude that peritoneal oxygenation is a promising technique that warrants further investigation and might be used in clinical settings in the future.

**Keywords:** rescue ventilation; peritoneal oxygenation; peritoneum; ventilation

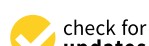



## 1. Introduction

Acute respiratory distress syndrome (ARDS) is an acute life-threatening clinical illness. It is a sequela of inflammatory lung injury and is clinically identified by hypoxia and poor lung compliance due to increased pulmonary vascular permeability. Patients who develop ARDS are very likely to require mechanical ventilation (MV) until the lungs recover and regain normal function and gas exchange capability [1,2].

ARDS was first described by Ashbaugh et al. in 1967 [3], and since then, our understanding of the pathogenesis has improved considerably. At the same time, treatment modalities have evolved but the mortality rate remains high, ranging from 27% for milder cases to 45% for the more severe [1,2]. ARDS is characterised by gas exchange impairment,

decreased lung compliance and increased pulmonary arterial pressure. This results from an alveolar injury, irrespective of the cause. It is followed by the release of pro-inflammatory cytokines that recruit neutrophils to the lungs and alveoli where they become activated and release toxic mediators that damage capillary endothelium and alveolar epithelium, leading to alveolar oedema [1,4].

Extracorporeal membrane oxygenation (ECMO) is a rescue therapy that can be used when patients with ARDS are no longer responding to conventional MV [5]. However, it is associated with considerable morbidity and procedural complexity [5,6]. The COVID-19 pandemic has highlighted that extrapulmonary ventilation, including accessible alternatives, is something we should look at more than ever.

Peritoneal membranous oxygenation was first described in 1934 by Singh [7,8]. This concept was derived from the experience with peritoneal dialysis and began with ventilating the peritoneal cavity with $O_2$. Since then, various techniques, mediums and oxygen carriers have been proposed. Although the peritoneum can serve as an avid gas exchange membrane, there are several different characteristics of the lung that must be taken into consideration to achieve peritoneal membranous oxygenation (Table 1). Comparatively to the lung, there is reduced capacity for gas-exchange including in surface area and volume, which must be considered when considering peritoneal oxygenation; however, in the diseased lung, the gas exchange capability is likewise considerably reduced [9,10].

**Table 1.** Comparison of perfusion characteristics of the lung and peritoneum. Adapted with permission from Legband et al. [9,10].

| Gas Exchange Locations | Surface Area ($m^2$) | Volume (L) | Blood Flow (L/min) | Proportion of Cardiac Output (%) | Membrane Thickness (µm) |
|---|---|---|---|---|---|
| Lung | 24–75 | 4–6 | 4.4–8.4 | 100 | 0.1–2.2 |
| Peritoneum | 1–2 | 2–3 | 1.4–2 | 17–30 | 24–500 |

## 2. Peritoneal Membranous Oxygenation Techniques

The concept of peritoneal membranous oxygenation was extrapolated from the method of peritoneal dialysis, which uses the peritoneum as a surface for gas exchange. Numerous techniques and mediums have been explored in several animal models and using a variety of delivery methods (Appendix A). Peritoneal oxygenation requires both a method to deliver a perfusion medium to the peritoneum in addition to providing a provision medium with significant oxygen-carrying and transfer capability.

### 2.1. Peritoneal Jet Ventilation

Peritoneal jet ventilation is a technique in which two intraperitoneally placed catheters for inflow and outflow are attached to a ventilator. Wang et al. used 36 swine models in their study to demonstrate the optimal frequency of ventilation cycles to maintain adequate tissue oxygenation [11]. The swine models underwent peritoneal ventilation after being anaesthetised and preoxygenated with 100% $O_2$. The swine models were then divided into groups and ventilated at different frequencies to demonstrate the optimal frequency at which prolonged apnoea could then be tolerated.

Overall, Wang et al. [11] demonstrated that the duration of safe apnoea increased with an increased frequency of jet ventilation, with the longest safe apnoea duration in the groups at a frequency of 80–120 times per minute. It should be noted that, as far as we know, this study is the first one of its kind to examine this technique.

### 2.2. Continuous Low-Pressure Oxygen Flow System in ARDS

Peritoneal oxygenation has been evaluated in an ARDS-swine model by Lemus et al. [12]. The study was conducted on four swine, which were invasively monitored for eight hours while under anaesthesia. Three laparoscopic trocars were placed into the peritoneal cavity

and a continuous oxygen flow of 5–6 L/min was maintained using a continuous low-pressure oxygen system (PEROX). After five hours, oleic acid was administered via the pulmonary circulation to induce ARDS, which was later confirmed by histopathology. PEROX maintained normal oxygenation indices both before and after the outset of ARDS; there was no statistically significant difference in oxygenation before or after ARDS [12]. Although this study would appear to demonstrate efficacy for the PEROX continuous oxygenation model, there is only limited methodology and reporting available, and it has not been replicated, so it should be treated with some caution.

### 2.3. Peritoneal Microbubble Oxygenation

Peritoneal microbubble oxygenation (PMO) is a novel technique that utilises the circulation of phospholipid-coated oxygen microbubbles (OMBs) in the peritoneal cavity. OMBs have a high oxygen-carrying capacity as they are 1% phospholipid, and the remaining oxygen is suspended in a saline solution. The major work on OMBs and their use in PMO were conducted by the University of Nebraska [13], with the initial concept being developed by Feshitan et al. [14]. A rat pneumothorax model was utilised, with five rats receiving intraperitoneal OMBs, five receiving oxygenated saline and six left untreated. The group that received the OMBs had a 100% two-hour survival rate, compared to the mean survival times of $15.5+/-8.1$ min and $16.0+/-4.8$ min for oxygenated saline and untreated groups, respectively. The difference in survival was statistically significant ($p < 0.004$), demonstrating the utility of OMBs in rats. After proving effectiveness in smaller rat models, larger rabbit models were utilised and efficacy was replicated with some success. In this study, Legband et al. used tracheal occlusion to induce hypoxia in rabbits. They discovered that when these rabbits received an intraperitoneal bolus of OMBs, their survival duration significantly increased from $6.6 \pm 0.6$ to $12.2 \pm 3.0$ min when compared to a control group ($p < 0.004$) [15].

Building on the work from the Nebraska team, a recent study by Fiala et al. has explored OMBs in a rat ARDS model [13]. ARDS was induced using inhalation of lipopolysaccharide (LPS) and confirmed on necropsy. Then, after the administration of LPS, the rats were divided into three groups: the control (no treatment) group, the group that received a saline bolus and the treatment arm that received 100 mL/kg OMBs as an intraperitoneal bolus every 12 h. After 48 h, the groups' rates of survival were 43%, 57% and 80%, respectively, with the OMB group showing increased survival over the other two groups ($p = 0.002$, $p = 0.009$). The treatment group also had higher peripheral blood oxygen saturation. Interestingly, the treatment group had a lower lung wet/dry ratio, indicating less pulmonary edema and better lung health [13]. Application of OMBs in larger animal models is needed to further demonstrate safety and efficacy.

### 2.4. Intraperitoneal Oxygenated Perfluorocarbon

Perfluorocarbons are organic fluorine-based inert compounds that are colourless and non-toxic. Perfluorocarbons were chosen as oxygen carriers because of their inert characteristics and excellent gas-dissolving function. For comparison, it has been shown that the $O_2$ and $CO_2$ solubility of perfluorocarbons is around 16 times and 4 times when compared to saline, respectively [16–20]. Given these properties, perfluorocarbon delivery via the peritoneum has been proposed as a method for intraperitoneal ventilation.

Carr et al. [16] used swine to demonstrate the feasibility of peritoneal ventilation with perfluorocarbons in large animal models. The study included 15 anaesthetised pigs and a state of hypoxia was induced to subatmospheric levels, down to an $FiO_2$ of 10%. A peritoneal ventilation circuit was created using large ECMO venous return catheters in the abdomen requiring laparotomy; they were secured in the upper left and right abdomen. The intraperitoneal oxygenation circuit was initiated and $FiO_2$ steadily increased. Peritoneal perfusion was performed in eight animals with oxygenated perfluorocarbon and seven animals with oxygenated saline. Oxygenated perfluorocarbon resulted in a higher average increase in $PaO_2$ than oxygenated saline, with a mean increase of 12.8 mm Hg (95% CI,

7.4–18.2 mm Hg; $p < 0.001$). The effects were statistically significant across the range of $FiO_2$ levels examined, with the most significant difference occurring at an $FiO_2$ of 14%. At 14% $FiO_2$, oxygenated saline had a $PaO_2$ of 39.4+/−5.0 mm Hg and 55.3+/−7.6 mm Hg using oxygenated perfluorocarbon. This corresponds to an increase in oxygen saturation from 73% to 89% and, therefore, to an oxygen saturation level that confers theoretical survivability. This was the first study to demonstrate the potential feasibility of ventilation in large animal models and it came to the conclusion that in order to achieve peritoneal oxygenation, a medium with a large oxygen capacity, such as perfluorocarbon, would be necessary. There were several limitations noted by the author, including challenges maintaining a stagnant volume of dwelling perfusate and the study only utilising extreme hypoxic states, which raises questions about whether the process would be useful for prolonged periods as an alternative to short to medium-term ventilation in a critical care setting [16]. They do; however, acknowledge that oxygenation through this method would be targeted toward those critically hypoxic patients who require heroic resuscitative methods.

*2.5. Perfusion with Oxygenated Red Blood Cells*

The use of oxygenated red blood cells as an intraperitoneal perfusion medium was assessed in canines. The canine models were anaesthetised and kept normoxic through MV with room air and then received either intraperitoneal circulation of saline, oxygenated saline or oxygenated human red blood cells. The group circulating red blood cells displayed increased levels of $PaO_2$, demonstrating oxygen diffusion across the peritoneal membrane, extrapolating from data obtained from peripheral arterial samples taken during anaesthesia. The medium was delivered through catheters directed towards the paracolic gutter, resulting in a modest increase in $PaO_2$. While oxygenation with oxygenated red blood cells may be feasible, this was assessed in normoxic models; therefore, its application to the critically ill is uncertain [21].

**3. Future Developments and Limitations**

While we believe peritoneal oxygenation is a promising technique, further development is needed before implementation on human participants. These should focus on both the delivery mechanism and the medium. Prior research on the use of perfluorocarbons as a red blood cell substitute has had mixed results in human trials outside of peritoneal oxygenation, with many either having safety or efficacy concerns when used intravenously. However, the research into perfluorocarbon usage in the peritoneal oxygenation space is promising and should warrant further investigation because the administration through the peritoneum may mitigate some of the issues experienced with an intravenous application. Although the efficacy currently appears promising and there have not been any significant issues identified with perfluorocarbons when administered intraperitoneally in animal models, further evaluation of safety needs to be conducted.

One of the difficulties in achieving extrapulmonary ventilation using alternative methods, such as ECMO, is the complexity and complications associated with the delivery mechanisms. Peritoneal oxygenation would need to be delivered using a simple and safe delivery mechanism to be a practical alternative. Currently, the mechanisms used in most of the animal models are highly invasive and could result in significant morbidity in a human setting. In addition, several had trouble maintaining a steady and reliable flow volume to the peritoneum. Before applying to humans, these issues need to be resolved in the animal setting.

Given the relative infancy of the technology, there are likely other safety concerns which have not yet arisen. However, we can extrapolate from issues affecting peritoneal dialysis and should be prepared to manage similar complications. These include infection, or more specifically, bacterial peritonitis through the introduction of microorganisms into the peritoneal cavity, misplacement of the delivery apparatus or damage to intraabdominal structures [22].

## 4. Conclusions

Peritoneal oxygenation as a method of providing extraperitoneal ventilation is a novel approach that has demonstrated some promising results. Further development of an oxygen delivery medium, such as the perfluorocarbons described, in addition to a simplified and less invasive delivery mechanism in animal models could permit future trials in human applications.

**Author Contributions:** Conceptualisation, K.H.A., D.L.M.; methodology, K.H.A., J.R.M.C.; investigation, K.H.A., J.R.M.C.; data curation, K.H.A., J.R.M.C.; writing—original draft preparation, J.R.M.C., K.H.A.; writing—review and editing, J.R.M.C., D.L.M.; supervision, D.L.M.; project administration, D.L.M. All authors have read and agreed to the published version of the manuscript.

**Funding:** This research received no external funding.

**Institutional Review Board Statement:** Not applicable.

**Informed Consent Statement:** Not applicable.

**Data Availability Statement:** Not applicable.

**Conflicts of Interest:** The authors declare no conflict of interest.

## Appendix A

**Table A1.** Summary of major published work describing peritoneal oxygenation.

| Author | Animal Model | Hypoxia Method | ARDS Method | Delivery Method | Outcome |
|---|---|---|---|---|---|
| Matsutani et al. (2008) [21] | Canine [$n = 18$] | Normoxia maintained through room air MV | N/A | Oxygenated red blood cells and oxygenated saline through peritoneal infusion | Improved systemic oxygenation in both treatment groups when compared to the control group ($p < 0.05$). |
| Carr et al. (2006) [16] | Swine [$n = 15$] | $FiO_2$ reduced to 10% | N/A | Oxygenated perfluorocarbons and oxygenated saline through peritoneal infusion | Mean increase of 12.8 mm Hg (95% CI, 7.4–18.2 mm Hg; $p < 0.001$) in oxygenated perfluorocarbon arm. |
| Wang et al. (2014) [11] | Swine [$n = 36$] | $FiO_2$ progressively reduced | N/A | Peritoneal jet ventilation | Duration of safe apnoea increased, an increased frequency of jet ventilation, the longest safe apnoea duration in the groups at a frequency of 80–120 times a minute. |
| Lemus et al. (2006) [12] | Swine [$n = 4$] | ARDS | Oleic acid administered to pulmonary circulation | PEROX | PEROX maintained normal oxygenation indices both prior to and following the outset of ARDS; there was no statistically significant difference in oxygenation prior to or following ARDS. |
| Feshitan et al. (2014) [14] | Rat [$n = 16$] | Pneumothorax | N/A | Phospholipid-coated OMBs via PMO given as an intraperitoneal infusion | Group receiving OMBs had a 100% two-hour survival, the oxygenated saline and those who were untreated had a mean survival of $15.5 \pm 8.1$ min and $16.0 \pm 4.8$ min, respectively ($p < 0.004$). |

**Table A1.** *Cont.*

| Author | Animal Model | Hypoxia Method | ARDS Method | Delivery Method | Outcome |
|---|---|---|---|---|---|
| Legband et al. (2015) [15] | Rabbit [$n = 19$] | Tracheal occlusion | N/A | OMBs via PMO delivered as an intraperitoneal bolus | Increase in the survival duration of those rabbits given an intraperitoneal bolus of OMBs from $6.6 \pm 0.6$ to $12.2 \pm 3.0$ min when compared to the control group ($p < 0.004$). |
| Fiala et al. (2020) [13] | Rat [$n = 23$] | ARDS | LPS inhalation | Phospholipid-coated OMBs via PMO given as a bolus | Improved survival of 37% after 48 h between rats given OMB bolus and the no treatment control with higher peripheral blood oxygen saturation. Lower lung wet/dry lung ratio in the OMB treatment group indicating less pulmonary edema. |

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
