# Peer review of "Peritoneal Oxygenation as a Novel Technique for Extrapulmonary Ventilation; A Review and Discussion of the Literature"

_arm, doi:10.3390/arm90060057_

Round 1

Reviewer 1 Report (Previous Reviewer 4)

Dear authors, I think that this corrected version is reasonable enough. I have no further comments.

Author Response

Thank you reviewer for the positive comments. 

Reviewer 2 Report (Previous Reviewer 2)

The revised manuscript is much improved. 

Author Response

Thank you to the reviewer for the positive feedback.

Reviewer 3 Report (Previous Reviewer 1)

The paper might be considered after minor revisions.

Extensive English improvements are needed, some passages are otherwise not clear.

Highlights should be reorganized, particularly passage in lines 17-20 is not clear.

Numerous typos are present, only for example line 12 “in” instead of “is”; line 18 please specify the subject; line 30 “on” instead of “and”; line 186 and 252 remove “;” line 262 “in” instead of “is”  

Keywords: please provide more specific keywords, please do not repeat the same keyword “ventilation” twice

Lines 62-64: please expand this passage. 

Table 1 is not readable. Moreover please provide a description of Table 1.

Author Response

Thank you to the reviewer for the feedback.

The paper might be considered after minor revisions.

Extensive English improvements are needed, some passages are otherwise not clear.

 - We have had the manuscript edited by a professional service. Thank you for the feedback.

Highlights should be reorganized, particularly passage in lines 17-20 is not clear.

 - We have reorganized and reworded the highlights. 

Numerous typos are present, only for example line 12 “in” instead of “is”; line 18 please specify the subject; line 30 “on” instead of “and”; line 186 and 252 remove “;” line 262 “in” instead of “is”  

 - Addressed with editor thank you

Keywords: please provide more specific keywords, please do not repeat the same keyword “ventilation” twice

 - We have altered and addressed this.

Lines 62-64: please expand this passage. 

 - The passage has been expanded to clarify meaning. 

Table 1 is not readable. Moreover please provide a description of Table 1.

 - Have redone table 1 to be readable within the manuscript document and a description. 

This manuscript is a resubmission of an earlier submission. The following is a list of the peer review reports and author responses from that submission.

Round 1

Reviewer 1 Report

This is a review focusing on the available literature on peritoneal oxygenation, a technique used for extrapulmonary ventilation. The techniques presented are all derived from animal experiments in which hypoxia was induced by applying several methods. The authors state that urgent studies on peritoneal oxygenation are needed, especially to explore its potential use in SARS-CoV-2 induced ARDS and hypoxia, for those cases when traditional mechanical ventilation fail to save the patients.

The review is quite well written, although language and paragraphs should be improved.

Themes treated are curious and engaging, however I opine that authors must avoid to suggest human application as they do in section 3 (Potential Use in COVID-19), unless their considerations are supported by evidence on specific experiments found in literature on humans, which authors do not cite in the text. It is methodologically not correct to conclude or suggest to do something for which no validated experiments and data are available and reproducible. Are the described suggestions derived from personal data or experience?

I suggest to reject the paper unless section 3 would be thoroughly revised.

In case authors would reformulate section 3, the article might be considered after major revisions:

1)    Some of the techniques are not sufficiently or clearly explained. In particular, paragraph 2.2 need to be expanded with more details.

2)    Authors should also add a paragraph on the potential limit of extrapulmonary ventilation. For example, Schmidt JA et al described that physiologically significant amounts of O2 cannot be delivered through peritoneal oxygenation because of the low transport coefficient of the peritoneum and small effective surface areas in the abdominal cavity (ASAIO Trans 1989;35(1):35-9).

3)    Typos: line 54 pag 2 patients instead of patient; line 102 pag 3 “oxygenated” add “saline”; 

4)    The title should be ameliorated. It is not clear that all the experiments presented are on animal models. It seems that the results described could be rapidly applicable to COVID-19 while this is not correct as no evidences are available on this topic.  

Author Response

In case authors would reformulate section 3, the article might be considered after major revisions:

1)    Some of the techniques are not sufficiently or clearly explained. In particular, paragraph 2.2 need to be expanded with more details.

We have amended those portions of section 2 and added clarification of techniques including in 2.2. 

2)    Authors should also add a paragraph on the potential limit of extrapulmonary ventilation. For example, Schmidt JA et al described that physiologically significant amounts of O2 cannot be delivered through peritoneal oxygenation because of the low transport coefficient of the peritoneum and small effective surface areas in the abdominal cavity (ASAIO Trans 1989;35(1):35-9).

We have added limitations in section 3 including in delivery method. We do believe more contemporary research including the swine studies we have discussed have demonstrated it may be possible to provide physiological oxygenation with techniques developed since the initial paper described in 1989. These should include the perfluorocarbons and OMB mediums.

3)    Typos: line 54 pag 2 patients instead of patient; line 102 pag 3 “oxygenated” add “saline”; 

Have addressed thank you. 

4)    The title should be ameliorated. It is not clear that all the experiments presented are on animal models. It seems that the results described could be rapidly applicable to COVID-19 while this is not correct as no evidences are available on this topic.  

We have altered the title and on retrospect after taking on the feedback from reviewers and reassessing the literature have modified our conclusion to more appropriately match where current research stands.

Reviewer 2 Report

The review brings us toward a new alternative method to improve oxygenation in ARDS via peritoneum.

Several elements that need to be improved are as follows:

- An image or diagram showing the role of the peritoneum for gas exchange may be provided.

- Information on page 3, lines 80-81 is non-specific and could potentially misinform the reader. The original article summarized that at 37 C, O2, and CO2, the solubility in perfluorocarbons is 50 mL and 200 mL per 100 mL of PFC, respectively. The solubility of O2 and CO2 is 3 and 57 ml per 100 ml of saline, respectively.

- Page 3, lines 88-89, please double check the interpretation of the data from the perspective of deoxygenation vs. oxygenation. Carr SR, et al. reported a drop of PaO2 in 15 intubated pigs. After rendering subatmospheric concentrations of inspired oxygen ranging from 18 to 10%, a drop in (baseline) mean PaO2 range of 65.9 ± 9.7 to 26.6 ± 2.8 mm Hg was found.

- Page 3, lines 101-102, the authors may have left out a word. Perhaps it should be “who received oxygenated saline.”

Additionally, more information on animal models until solid evidence confirmation the benefit is needed. It is too soon to conclude this technique could provide to an actual human. So, the paragraph about the potential use of peritoneal oxygenation in COVID-19 might not yet be possible.

Author Response

  • An image or diagram showing the role of the peritoneum for gas exchange may be provided.
  • Information on page 3, lines 80-81 is non-specific and could potentially misinform the reader. The original article summarized that at 37 C, O2, and CO2, the solubility in perfluorocarbons is 50 mL and 200 mL per 100 mL of PFC, respectively. The solubility of O2 and CO2 is 3 and 57 ml per 100 ml of saline, respectively. 
    • We have restated and simplified this statement and converted to a more approachable ratio. 
  • - Page 3, lines 88-89, please double check the interpretation of the data from the perspective of deoxygenation vs. oxygenation. Carr SR, et al. reported a drop of PaO2 in 15 intubated pigs. After rendering subatmospheric concentrations of inspired oxygen ranging from 18 to 10%, a drop in (baseline) mean PaO2 range of 65.9 ± 9.7 to 26.6 ± 2.8 mm Hg was found.
    • We have altered and simplified our summary of this study to more accurately convey the purpose of the study.
  • - Page 3, lines 101-102, the authors may have left out a word. Perhaps it should be “who received oxygenated saline.”
    • Grammar has been corrected.
  • Additionally, more information on animal models until solid evidence confirmation the benefit is needed. It is too soon to conclude this technique could provide to an actual human. So, the paragraph about the potential use of peritoneal oxygenation in COVID-19 might not yet be possible.
    • We agree with the feedback of the reviewers and have modified our conclusion based upon the currently available technology and research. 

Reviewer 3 Report

The purpose of this review is to summarize the efficacy of peritoneal oxygenation and to propose further investigation of its role in severe ARDS due to COVID-19.

In my point of view this is an interesting review, though authors haven’t signified the potential risks from peritoneal oxygenation by any type.

I suggest only minor comments to the authors:

1.       The text needs correction of double spaces, noted throughout the text.

2.       The paragraph in the lines 49-52 needs some editing to be clearer, as well as the lines 102-107, and 112-115.

3.       Line 54 : I propose patient to be replaced with patients  

4.       Line 72 54:  I propose concluding to be replaced with concluded

5.        The word apnoea should be written with one style in the whole text , while authors use both apnoea and apnea

6.       Line 89, and 95: the ranges of oxygenations are not showed correctly.

7.       Line 101: that is not needed

8.       Line 171: man should be replaced by human

Additional comments:

1.           Authors should specify and detail the potential risks from the technics described regarding peritoneal oxygenation.

2.           The tittle of the review should be modified as it should highlight that peritoneal oxygenation is an experimental method tested in animal only and not in human, so far.

3.           The section 3 needs revision and more detailed description.

Author Response

  1. The text needs correction of double spaces, noted throughout the text.
  2. The paragraph in the lines 49-52 needs some editing to be clearer, as well as the lines 102-107, and 112-115.
  3. Line 54 : I propose patient to be replaced with patients  
  4. Line 72 54:  I propose concluding to be replaced with concluded
  5.  The word apnoea should be written with one style in the whole text , while authors use both apnoea and apnea
  6. Line 89, and 95: the ranges of oxygenations are not showed correctly.
  7. Line 101: that is not needed
  8. Line 171: man should be replaced by human

We have addressed those grammatical issues thank you.

Additional comments:

  1. Authors should specify and detail the potential risks from the technics described regarding peritoneal oxygenation.

We have added an additional paragraph detailing potential risks in addition to a section regarding limitations of the technique.

  1. The tittle of the review should be modified as it should highlight that peritoneal oxygenation is an experimental method tested in animal only and not in human, so far.

We agree and have altered the title and abstract to better reflect the current status of research.

  1. The section 3 needs revision and more detailed description.

We have rewritten both section 2 and 3 to provide more detail and a more a narrative review. 

Reviewer 4 Report

This is a review of the literature on animal models, with the aim of use of peritoneal oxygenation on critically unwell hypoxic patients, as a promising alternative to rescue ventilation.

Comments:

·         I found this article very easy to follow, although there were some syntax errors.

·         Peritoneal oxygenation is presented by the authors to be a simple and cost-effective treatment in extreme hypoxia, in resource-poor climates. However most of these alternative methods of oxygenation suggested in animals may not be available in resource poor climates, as well.

·         Why not intrapulmonary oxygenation with perfluorocarbons? To my knowledge, previews studies with perfluorocarbons in human adults  were led to early termination, because these substances were proven not to be inert.

Author Response

 I found this article very easy to follow, although there were some syntax errors.

  • Peritoneal oxygenation is presented by the authors to be a simple and cost-effective treatment in extreme hypoxia, in resource-poor climates. However most of these alternative methods of oxygenation suggested in animals may not be available in resource poor climates, as well.

We agree and in retrospect given the infancy of the technique we have removed that comment although do believe that there is the potential to offer some procedural simplicity when compared to current other extrapulmonary ventilation techniques i.e. ECMO. 

  • Why not intrapulmonary oxygenation with perfluorocarbons? To my knowledge, previews studies with perfluorocarbons in human adults  were led to early termination, because these substances were proven not to be inert.

We wanted to focus on extrapulmonary ventilation in this review and have added some limitations to discussion about perfluorocarbons to reflect some of the safety issues in a number of areas. There is some more contemporary research into total liquid ventilation with perfluorocarbons which demonstrated safety in primates such as by Kohlhauer et al 2020, however we believe there is considerable work to do in perfluorocarbons.